# The Kinetics of Aragonite Formation from Solution via Amorphous Calcium Carbonate

**DOI:** 10.3390/nano12234151

**Published:** 2022-11-23

**Authors:** Simon M. Clark, Vili Grigorova, Bruno Colas, Tamim A. Darwish, Kathleen Wood, Joerg Neuefeind, Dorrit E. Jacob

**Affiliations:** 1School of Engineering, Faculty of Science and Engineering, Macquarie University, North Macquarie Park, Shellharbour, NSW 2109, Australia; 2Australian Centre for Neutron Scattering, Australian Nuclear Science and Technology Organisation, Locked Bag 2001, Kirrawee DC, Sydney, NSW 2232, Australia; 3National Deuteration Facility, Australian Nuclear Science and Technology Organisation, Kirrawee DC, Sydney, NSW 2232, Australia; 4Spallation Neutron Source, Oak Ridge National Laboratory, Oak Ridge, TN 37831, USA; 5Research School of Earth Sciences, The Australian National University, Canberra, ACT 2600, Australia

**Keywords:** neutron scattering, X-ray diffraction, Small Angle Neutron Scattering, Laser-Ablation Inductively Coupled Plasma Mass Spectrometry, themo gravimetric analysis, NOMAD, QUOKKA, ACC, aragonite, Mg/Ca

## Abstract

Magnesium doped Amorphous Calcium Carbonate was synthesised from precursor solutions containing varying amounts of calcium, magnesium, H_2_O and D_2_O. The Mg/Ca ratio in the resultant Amorphous Calcium Carbonate was found to vary linearly with the Mg/Ca ratio in the precursor solution. All samples crystallised as aragonite. No Mg was found in the final aragonite crystals. Changes in the Mg to Ca ratio were found to only marginally effect nucleation rates but strongly effect crystal growth rates. These results are consistent with a dissolution-reprecipitation model for aragonite formation via an Amorphous Calcium Carbonate intermediate.

## 1. Introduction

Amorphous calcium carbonate (ACC) plays a key role in the early stages of biomineralisation as a precursor to crystalline calcium carbonate phases (e.g., [1]) and is a model system in the development of biomimetic materials. Many organisms take advantage of the mouldable character of ACC in the formation of their intricate shells and skeletons. ACC also seems to lower the activation energy for nucleation of subsequent crystalline carbonates due to lower surface energies [2] enabling an ambient temperature non-classical crystallisation pathway [3]. It is known that ACC is polyamorphous with different water content, local order and mode of formation [4,5], and it was recently shown that ACC is best described as a nano-fluid [6]. Understanding the mechanism by which these nanoparticles self-assemble is essential if we are to understand the effect of climate change on marine life and how well we are presently using the fossil record to constrain climate change models.

In the laboratory ACC converts to the crystalline carbonate phases in a matter of minutes although, depending on the aqueous fluid composition and any additives [7], its stability can be increased allowing ACC to be preserved for weeks to years [8,9]. The additives most commonly used in the laboratory to prolong ACC lifetime are organic macromolecules, such as polyacrylic acid [10,11] and magnesium [11,12,13,14]. In natural biominerals, in addition to Mg, a large range of organic macromolecules can exert a stabilising effect on ACC, which can extend the stability of ACC almost indefinitely [15]. Mg-ACC usually transforms into calcite at low Mg content [16,17,18,19], but at high Mg content transforms into aragonite [17,18].

Non-classical nucleation and growth models often provide a better description of biomineralization kinetic data than classical crystallisation models [3]. In this context, two models have been used to describe the transformation mechanism from ACC to aragonite or calcite in non-classical crystallisation: a solid-state transformation, where ACC dehydrates and crystallises into a calcium carbonate polymorph directly in the solid phase [1,20]; or a dissolution-reprecipitation process [20,21,22], where ACC dissolves and crystalline calcium carbonate precipitates from this solution. In the laboratory, ACC usually transforms via dissolution precipitation which is typically associated with significant morphological changes to the texture of the solid phase [20,23,24]. In vivo, preservation of intricate hierarchical structural details [25] during stepwise crystallisation seems to contradict the findings in the laboratory, but supports either solid-state transformation, or very localised dissolution-reprecipitation processes or perhaps a combination of both [23,26].

To try to distinguish between these two models, we measured the rate of formation of crystalline aragonite from ACC in the laboratory for a range of Mg/Ca ratio and H_2_O/D_2_O compositions.

## 2. Methodology

### 2.1. Sample Preparation

A set of ACC samples with a range of Mg/Ca ratios and water contents were synthesised with Mg/Ca ratios expected to give measurable formation and crystallisation rates. A commonly used synthesis procedure was followed [27] and modified with or without an additional freeze-drying step to allow variation of water content. Stock 0.02 M solutions of Na_2_CO_3_ and 0.02 M solutions made by dissolving various mixtures of solid CaCl_2_ and MgCl_2_ in water or heavy water to give solutions with a range of Mg/Ca content, were prepared and stored at 4 °C. About 100 mL of both the stock Na_2_CO_3_ and CaCl_2_/MgCl_2_ solutions were then quickly mixed together in a 250 mL bottle held at 4C while stirring using with a magnetic stirrer. Almost instantly the solutions were mixed a white precipitate formed. The white precipitate was filtered and washed with acetone and then either stored under nitrogen at room temperature or freeze-dried for one hour before being stored under nitrogen at room temperature. Stock solutions were prepared using H_2_O, D_2_O and mixtures of H_2_O and D_2_O as the solvent in order to vary the solvent chemistry and probe the effect of solvent on sample structure, stability, and crystallisation products.

### 2.2. Analysis Techniques

Each ACC sample synthesised was characterised using Thermal Gravimetric Analysis (TGA), to determine the amount of water present, Laser-Ablation Inductively Coupled Plasma Mass Spectrometry (LA-ICPMS), to determine the Mg/Ca ratio, Neutron scattering to confirm the LA-ICPMS measurements, and X-ray diffraction (XRD) to confirm that there were no crystalline phases present in the sample. Each of the carbonates crystallised from our ACC samples were characterised using Raman Spectrometry and powder X-ray Diffraction to determine which calcium carbonate polymorph was present, Scanning Electron Microscopy to provide visual evidence of sample composition, and a second powder X-ray diffraction measurement incorporating an internal standard to provide accurate crystalline phase lattice parameters. Finally, time resolved Small Angle Neutron Scattering (SANS) was used to measure the rate of formation of ACC from solution, and time resolved powder X-ray diffraction was used to determine the rate of formation of crystalline carbonate phases upon transformation of the ACC samples.

The TGA measurements utilised a TA Instruments TGA 2050. The sample temperature was increased from 20 °C to 800 °C, at a rate of 10 °C/min in a N_2_ atmosphere while measuring the weight of the sample. For the LA-ICPMS we used a Photon Machines G2 Excimer laser system (wavelength 193 nm) coupled with an Agilent 7700 quadrupole ICP-MS. ACC samples were dried at 100 °C for 36 h, pressed into pellets and measured with laser spot size of 50 µm and pulse rates of 5 Hz (energy density 5.42 J/cm^2^). Backgrounds were measured for 30 s followed by 40 s of sample ablation. NIST SRM (US National Institute of Standard and Technology Standard Reference Material) 610 glass was used as the external standard with ^43^Ca as the internal standard. The mg for NIST SRM 610 reported in the Geo- ReM database was used as the “true” concentration in this reference glass [28]. Neutron scattering data were collected using the NOMAD instrument at the Spallation Neutron Source, Oak Ridge National Laboratory, USA. Samples synthesized using 100% D_2_O to reduce background noise from inelastic scattering of neutrons from protons, were loaded into 5 mm diameter quartz capillary tubes inside a glove box under a nitrogen atmosphere, sealed and quickly transported to the instrument. Scattering data were collected in 30 min frames at room temperature in an argon atmosphere for a total of 1.5 h. The standard instrument data reduction software was used to normalize the data and for background subtraction. XRD patterns were collected using a PANalytical Expert Pro MPD (Multi-Purpose Diffractometer) with a Copper tube (45 kV, 40 mA), Soller slits (0.04 rad.) and fixed incident beam mask (15 mm). The instrument zero point was calibrated using the NIST 640c Si standard. The samples were placed on zero background sample holders, and the wavelength used was CuKα (λ = 1.54 Å). For the accurate lattice parameter measurements appropriate amounts of the NIST, Silicon Powder, Standard Reference Material^®^ 640c was mixed with each sample to increase the accuracy of the measurement. Unit cell parameters were then determined using the Profex software package [29]. Series of powder XRD patterns were also collected from ACC samples using the same system at ambient temperature while they transformed into crystalline carbonate phases. Diffraction patterns were collected over the 10° to 50° 2θ range every 30 min for 24 to 48 h for a range of initial conditions. Peak intensities were determined from these spectra as a function of time and converted into a measure of the degree of reaction for subsequent analysis. Raman spectra were collected using a Horiba LABRAM HR Evolution confocal laser Raman spectrometer. The spectrometer used a red excitation laser (λ = 633 nm) and a 1800 gr/mm grating. Before collecting data from each sample, the spectrometer was calibrated using the Rayleigh line and a Si-wafer with a sharp peak at 520 cm^−1^. A JEOL JSM- 6480 LA Scanning Electron Microscope (SEM) was used to obtain SEM micrographs from gold coated samples. SANS data were collected using the QUOKKA instrument [30] at the Australian Centre for Neutron Scattering, Australian Nuclear Science and Technology Organisation, Lucas Heights, Australia. Samples were placed in quartz Hellma cells mounted in a thermally controlled sample changer. The neutron path length through the sample was 2 mm and the cell volume 600 µL. For each measurement, the solutions made from Na_2_CO_3_, CaCl_2_ and MgCl_2_ were mixed in the cells minimising any lag between mixing and the start of data collection. All solutions were made with only D_2_O, because samples with H_2_O were found to give too large a background signal due to incoherent scattering from protons. The source was set to a wavelength of λ = 5 Å, and the detector placed 8 m from the sample. Samples with Mg/Ca ratios of 0.03, 0.2 and 0.26 were studied at ambient temperature to simulate in vivo conditions as closely as possible.

## 3. Results

### 3.1. Characterisation of the Amorphous Phase

The main aims of the ACC characterisation were to check that the samples had not transformed to a crystalline carbonate phase, to determine the Mg/Ca ratios and the amount of water.

#### 3.1.1. Thermal Gravimetric Analysis

A typical TGA pattern is shown in Appendix A. The weight loss at around 100 °C was attributed to loss of surface water, the small weight loss around 350 °C was attributed to the loss of included water and the weight loss around 700 °C was attributed to the loss of carbon dioxide in keeping with previous studies [16,19]. The stoichiometric water loss was calculated from this weight loss assuming a sample formula of (Ca_1−x_Mg_x_)CO_3_·nH_2_O. The results are summarised in Appendix A. Freeze-dried samples were found to contain about 1.18 molecules of water per formula unit, and filtered samples were found to contain about 2.69 molecules of water. This was independent of the Mg/Ca ratio or whether the solvent contained H_2_O, D_2_O or a mixture of the two (Appendix A). These values are comparable with those found previously, such as n = 1.42–1.63 [19], and n = 0.98–1.4 [16]. Freeze drying can be seen to have the effect of reducing the amount of water percent in an ACC sample.

#### 3.1.2. X-ray Diffraction

XRD patterns of the freshly synthesised ACC samples were found to contain no Bragg peaks. We observed only broad humps indicative of an amorphous or nano-crystalline phase with two diffuse maxima at 32° (d = 2.8 Å) and 46° (d = 2.0 Å) 2θ (Appendix A) consistent with those previously measured from ACC samples [16].

#### 3.1.3. Laser-Ablation Inductively Coupled Plasma Mass Spectrometry

Representative measured Mg/Ca values are contained in Appendix A. In Figure 1 the Mg/Ca ratios determined using LA-ICPMS are plotted against the Mg/Ca ratios of the solutions from which the ACC samples formed. Comparing data from this study (blue spots) with data from other studies [23,30,31,32,33] (Figure 1) shows a linear correlation with the Mg/Ca ratio in solution, with an apparent distribution coefficient (K_d_ = [(Mg/Ca)_ACC_/(Mg/Ca)_solution_]) of 0.12. The exact relationship seems to be highly dependent upon the reported study with our data in the same range as data from two previous measurements [23,34]. Further reference to Mg/Ca ratios in this manuscript will refer to the Mg/Ca ratio of the solid sample and not the nascent solution. Potentially, the measured Mg/Ca ratio in the solid sample could be due to variations in the washing and drying protocols with some measurement signal coming from fluid still adhering to solid particles. To investigate this possibility, we also used neutron scattering to determine Mg/Ca ratios.

#### 3.1.4. Neutron Scattering

A previous study showed that scattering from an ACC sample containing only Ca has a peak in the real space scattering intensity at G(r) ≈ 2.3 Å, due to Ca-O correlations. In comparison, an ACC sample containing only Mg has a peak in the real space scattering intensity at G(r) ≈ 2.1 Å. ACC samples with mixtures of Ca and Mg have both peaks with the positions of these peaks varying linearly between the two solid solution end members (Appendix A). This provides a methodology for the measurement of Mg/Ca ratio in an ACC sample free from any remnant liquid phase contamination. Neutron scattering patterns of six samples containing different Mg/Ca proportions, x, given by the general formula: (Ca_1−x_Mg_x_)CO_3_·nH_2_O, are shown in Appendix A. The sample with Ca only contains one peak at ~2.4 Å due to Ca-O correlations, while the samples made from mixtures of Mg and Ca contain peaks at ~2.2 Å and ~2.4 Å due to Mg-O and Ca-O correlations, respectively. This observation demonstrates that Mg is indeed incorporated in the ACC atomic structure and is not a measurement artefact.

The positions of these two peaks are presented in Figure 2 as a function of Mg/Ca ratio together with data from a previous study [31]. The two datasets are seen to be well correlated. The peak due to Ca-O correlations at ~2.4 Å does not change with Mg/Ca ratio while the peak due to Mg-O correlations is seen to vary from ~2.2 Å at low Mg content to ~2.1 Å at 100% Mg content. These data confirm that the LA-ICPMS measurements provide an accurate determination of the Mg/Ca ratios in our samples. The data also demonstrate that comparison of the positions of the Ca-O and Mg-O correlation peaks from neutron scattering data provides an accurate method for the determination of Mg/Ca ratios with no need for any calibrations using standards.

The background intensity of the neutron scattering patterns can be used to estimate the H_2_O/D_2_O ratios in our samples by use of a calibration curve made from known pure H_2_O/D_2_O mixtures (Appendix A). This leads us to conclude that there is less than 10% H_2_O in our fully deuterated samples and, thus, more than 90% D_2_O. This provides an error estimate for our determined H_2_O/D_2_O ratios of about 5% due to contamination related to the highly hygroscopic nature of D_2_O. It translates into an uncertainty in the number of water molecules per formula unit of about ±0.06 for the freeze-dried samples and ±0.13 for the filtered samples. Comparing these uncertainties with the range of measured values using TGA of 0.9 to 1.37 for the freeze-dried samples and 2.12 to 3.46 for the filtered samples, it implies that the effect of freeze-drying vs. filtering is negligible.

### 3.2. Characterisation of the Crystalline Phases

The main aim of the characterisation of the crystalline phases was to determine which carbonate polytypes formed and their Mg/Ca ratios.

#### 3.2.1. Raman Spectroscopy

Raman spectra collected from our samples were compared to spectra we collected from reference samples of pure aragonite and calcite (Appendix A) Comparison of these spectra showed that the crystallised sample consisted solely of aragonite. This was confirmed by comparison with reference data from [35,36] and the RRUFF^TM^ database [37]. The spectra were found to contain three peak groupings: the ν_1_ symmetric stretching of the carbonate groups at 1085 cm^−1^; the ν_4_ in-plane bending of calcite at 711 cm^−1^, and the ν_4_ in-plane bending of aragonite at 706 cm^−1^ and 701 cm^−1^ and the lattice modes of calcite at 155 cm^−1^ and 281 cm^−1^, and the lattice modes of aragonite at 206 cm^−1^ and 155 cm^−1^. All crystallised samples measured consisted solely of the aragonite phase.

#### 3.2.2. X-ray Diffraction

XRD patterns from the crystalline phases produced from our ACC samples were fitted using the *Profex* (Solothurn, Switzerland) software [29] and were found to contain only reflections due to aragonite (Appendix A) showing that only aragonite crystals formed in agreement with our Raman data.

#### 3.2.3. Scanning Electron Microscopy

A representative SEM image is contained in the Appendix A. Clusters of tablet-shaped aragonite crystals can be seen. These are surrounded by xenomorphic, rounded grains that we interpret as ACC particles. The aragonite tablets are of similar size and shape as ones observed in nacre, 0.5 µm thick, 5–15 µm in diameter polygons [38]. This supports the conclusions of both Raman and XRD measurements that only aragonite is present in the crystallised samples.

#### 3.2.4. High-Precision X-ray Diffraction

Accurate determination of the Mg/Ca ratio in carbonates crystallised from ACC is difficult due to the presence of ACC particles on crystal faces (Appendix A) and in between the carbonate platelets making up the crystalline phase. The unit cell volume of carbonate phases is highly sensitive to any incorporation of other ions such as magnesium and should prove an accurate method for determining the Mg/Ca ratio. The unit cell volume of aragonite crystallised from our ACC as a function of Mg/Ca ratio in the precursor ACC is shown in Figure 3. The volume is seen to show only a small change from the volume of pure calcium containing aragonite with Mg/Ca ratio. How significant this small change is in the absence of literature aragonite unit cell volume data for varying Mg/Ca ratios can be estimated by comparison of volume change with substitution of other similar atoms. Volumes calculated for substitution of Ca by Sr in the aragonite (CaCO_3_) to strontianite (SrCO_3_) solid solution series [39] are also plotted in Figure 3. It was observed that there is an increase in unit cell volume with increasing amounts of Sr which is probably due to the Sr atom being bigger (in a crystal 1.32 Å) than a Ca atom (1.14 Å). In a similar manner, the unit cell volume of aragonite on substitution of calcium (1.14 Å) with magnesium (0.86 Å) was calculated the result of which is also plotted in Figure 3. Our conclusion is that there is no significant substitution of calcium by magnesium in the crystalline aragonite samples although it was previously incorporated in the ACC before transformation to aragonite.

### 3.3. Kinetics of Particle Growth and Crystallisation of Calcium Carbonate

SANS is the most suitable method for measuring the kinetics of ACC formation given the previously observed length scales of ACC particles. For determining the kinetics of aragonite formation time resolved powder X-ray diffraction is more appropriate given the crystalline nature of the product phase.

#### 3.3.1. Small-Angle Neutron Scattering

A time resolved SANS measurement determines scattering intensity as a function of momentum transfer for particles of nanometre to micro size in a series of time slices. The shape and slope of each of these scattering curves depends on the size and shape of the particles in the sample. A typical series of SANS spectra collected during the formation of one of our ACC samples is shown in Appendix A together with the extracted degree of formation curve and determined rate values. The data were fitted with two models; the first one was a simple spherical particle model, that considers all particles to be spheres and fits the radius to the data; and a Porod-Guinier model, that has a shape parameter as well as a size parameter using the SasView software package [40]. Both models gave similar results consistent with a spherical particle shape. ACC particles were found to start with an average radius of around 35 nm which grew to around 60 nm after 4–5 h for the samples with Mg/Ca ratios of 0.03 and 0.2. ACC particles with a Mg/Ca ratio of 0.26 also started with a radius of about 35 nm which remained constant for the duration of the measurement. A summary of particle size evolution under varying conditions is shown in Figure 4. The height of the peak in the SANS patterns was used to estimate the amount of ACC particles of each size as a function of time. These data were then fit with the Avrami equation [5]. This gave an estimate of the rate of formation of the initial 35 nm particles (rate = 0.17 h^−1^ for a Mg/Ca ratio of 0.26, rate = 0.27 h^−1^ for a Mg/Ca ratio of 0.2 and rate = 0.3 h^−1^ for a Mg/Ca ratio of 0.03) and the rate of transformation from 35 to 60 nm particles (rate = 2.8 h^−1^ for a Mg/Ca ratio of 0.03 and rate = 0.9 h^−1^ for a Mg/Ca ratio of 0.2 (Figure 5). The Mg/Ca ratio of 0.26 sample did not show any growth during the measurement period.

#### 3.3.2. X-ray Diffraction

A representative series of powder diffraction patterns are presented in Appendix A. The aragonite (111) and (221) peaks are seen to grow out of the background due to nano-crystalline calcium carbonate. The intensity of the (111) peaks were used to determine the degree of reaction for each series of XRD patterns which was then used with the Avrami equation [5] to give the corresponding reaction rate Figure 6). Broad amorphous scattering was observed but no sharp diffraction signal from a crystalline product within the time frame of our measurements when the ACC precursor had an Mg/Ca ration of above about 0.14. Some samples with very high Mg/Ca ratios were found to be still amorphous many months after they were synthesised.

## 4. Discussion and Conclusions

All samples of ACC synthesised during this study crystallised into aragonite as confirmed by XRD and Raman data. After the completion of crystallisation, a small amount of remnant ACC particles were observed on aragonite crystals, consistent with observations in natural systems such as mollusc shells [41,42]. Mg was found to be present in the nano-crystalline ACC atomic structure but not in the crystalline aragonite atomic structure. This suggests that Mg was lost during the transformation of ACC into aragonite. The lost Mg atoms most likely were incorporated into the remnant ACC particles observed on the surface of the aragonite crystals in our SEM images, stabilising this portion of the ACC to allow it to exist for extended periods of time as observed in ACC cortices around crystalline nacre tablets in mollusc shells [43,44].

The redistribution of Mg upon transformation of ACC into aragonite observed here requires transport of ions from the ACC nano-particles forming the crystalline phase, and thus, is consistent with a dissolution-reprecipitation process and discounts a pathway of solid-state transformation [3]. A schematic representation of the transformation process from solution through ACC to crystalline aragonite consistent with the data collected in this work is contained in Figure 7. We observe three steps:Precipitation of ACC from solution forming 35 nm particles: reaction R_1_.The 35 nm particles grow into 70 nm particles: reaction R_2_.Formation of aragonite crystals by localised dissolution and reprecipitation: reaction R_3_.

This is broadly in line with previous observations, for example: the precipitation of ACC particles of 35–40 nm [5], the observation of 100 nm particles in amorphous and crystal phases [42,45], 50–60 nm crystals [46], and 100 nm silica-coated ACC particles with 20% water [47]. Electron microscopy [48] as revealed the evolution of particles prior to step 1 above, with ACC particles growing from ~1 nm to ~20 nm in a range of a few seconds. Prenucleation clusters of size ~1 nm are thought to precede the nanoparticles of ~30 nm [49].

Values of the rates for step R_1_ were found to be largely independent of Mg/Ca ratio while the Mg/Ca ratio had a very large effect on the rates for steps R_2_ and R_3_. The change in rate for steps R_2_ and R_3_ with Mg/Ca ratio were found to be very similar. From a classical nucleation and growth perspective this might imply that the addition of magnesium mainly effects particle growth and has little effect on nucleation. This might explain why biomineralization predominantly involves nanoparticles since the nucleation rate is largely unregulated while the growth rate can be controlled.

All ACC samples were found to contain magnesium, but all the crystalline aragonite products were found not to contain any magnesium. The high rate for crystallisation observed suggests that this process does not involve diffusion in the solid state. This suggests that our results would favour a dissolution and reprecipitation model for ACC transformation to aragonite over a solid-state transformation model.

In this case, the nanoparticles were found to not simply orientate and attach to form larger structures but to dissolve and recrystallize as part of the self-assembly process. The bigger question we then ask is to what extent is this mechanism employed in self-assembly for both biological and synthetic systems.

## Figures and Tables

**Figure 1 nanomaterials-12-04151-f001:**
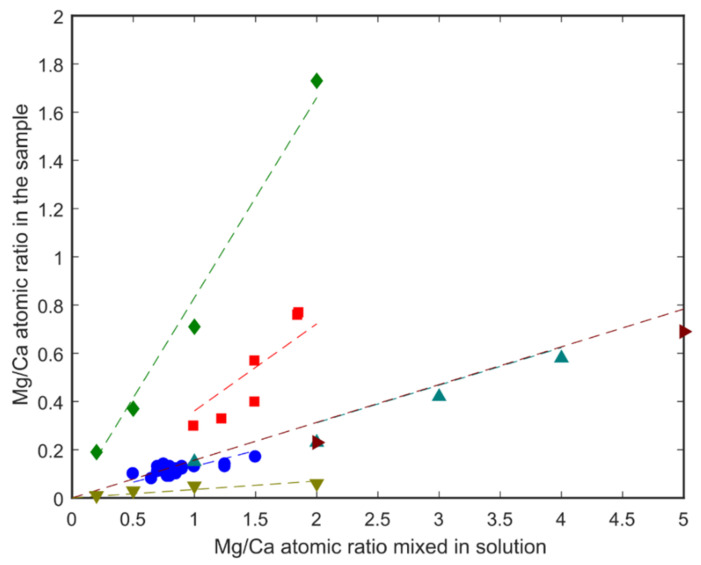
Comparison of Mg/Ca ratios measured in ACC samples to Mg/Ca ratios of the solutions from which they formed for samples: 
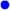
 from this study, 

 mixed with Formamide [33], 

 [31], 
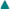
 [32], 
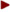
 [23], 
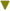
 with water [33]; the dashed lines are linear fits to the data points of corresponding colour.

**Figure 2 nanomaterials-12-04151-f002:**
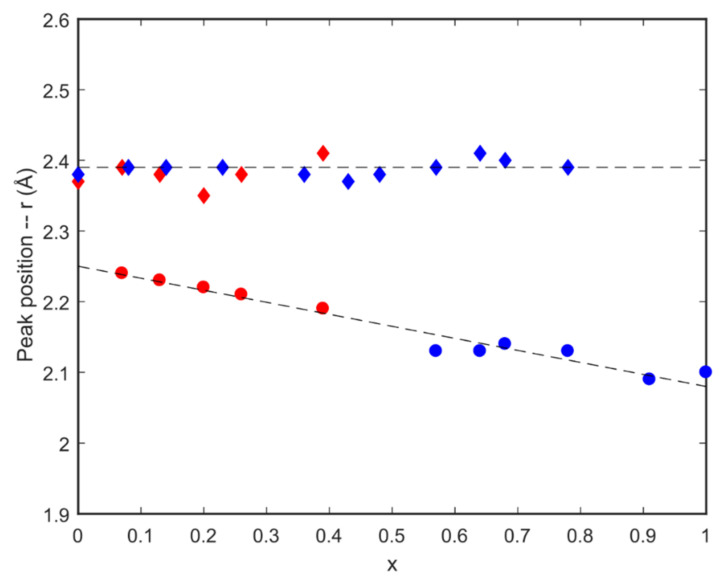
Peak positions of the first Mg-O and Ca-O correlations, taken from Appendix A, Where x is the proportion of magnesium in the sample as given by: (Ca_1−x_ Mg_x_)CO_3_. The data shown as red markers are from this study (Appendix A), and the blue markers from [31] (Appendix A). Filled diamond symbols denote the first Ca-O correlation peak around 2.4 Å; Filled circle symbols denote the first Mg-O correlation peak around 2.1–2.2 Å.

**Figure 3 nanomaterials-12-04151-f003:**
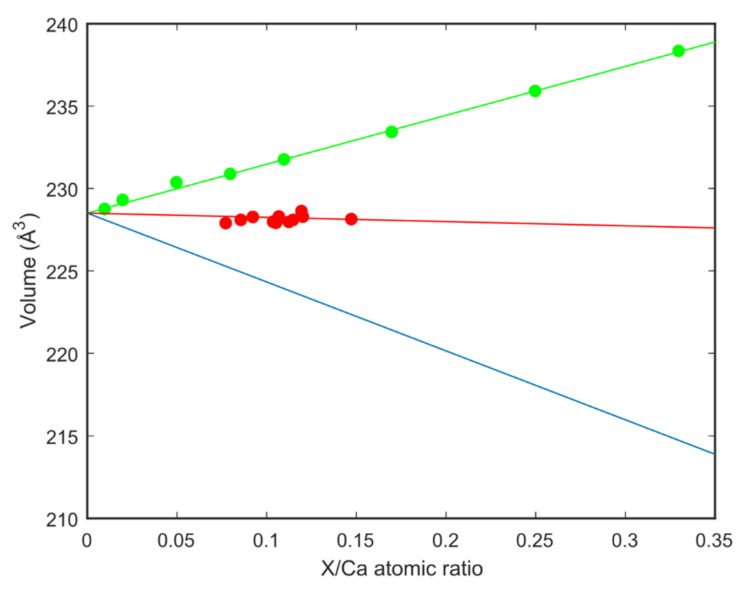
Data from this study compared to the unit cell volume change of the lattice estimated for substitution of Sr and Mg to Ca. X represents either Sr or Mg. 
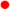
 Data from this study with the linear fit of the data points; 
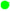
 Strontianite (SrCO3)-aragonite (CaCO3) solid solutions from [39] with the linear fit of the data points; 

 Estimate for Mg substituting Ca in the aragonite.

**Figure 4 nanomaterials-12-04151-f004:**
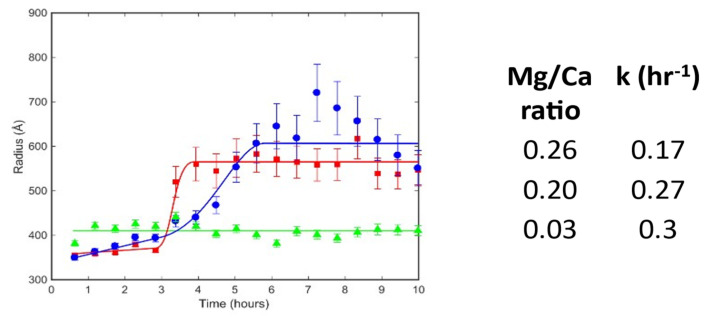
Evolution of the particle size in the samples analysed at 20 °C (ambient) with time. 
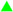
 is for the sample with Mg/Ca = 0.26; 
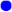
 is for the sample with Mg/Ca = 0.20; and 

 is for the sample with Mg/Ca = 0.03. The corresponding lines are guides for the eye. The table contains estimates of the rate of transformation between the 35 nm and nm particles for three Mg/Ca ratios.

**Figure 5 nanomaterials-12-04151-f005:**
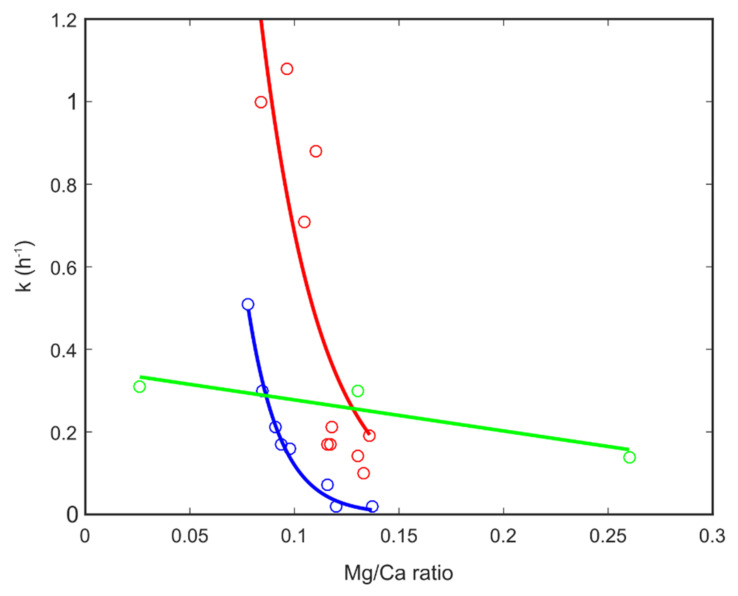
Reaction rates for the formation of initial ACC particles (green symbols and line) and subsequent formation of aragonite crystals (red symbols and line for formation in H_2_O and blue symbols and lines formation in D_2_O). The red and blue lines are exponential fits to the corresponding symbols, their equations are $y=22.87\;e^{-35.11\;x}$ for the red line and $y=76.38\;e^{-64.57\;x}$ for the blue line. The green line is a linear fit to the corresponding symbols giving: y=−0.75 x+0.35.

**Figure 6 nanomaterials-12-04151-f006:**
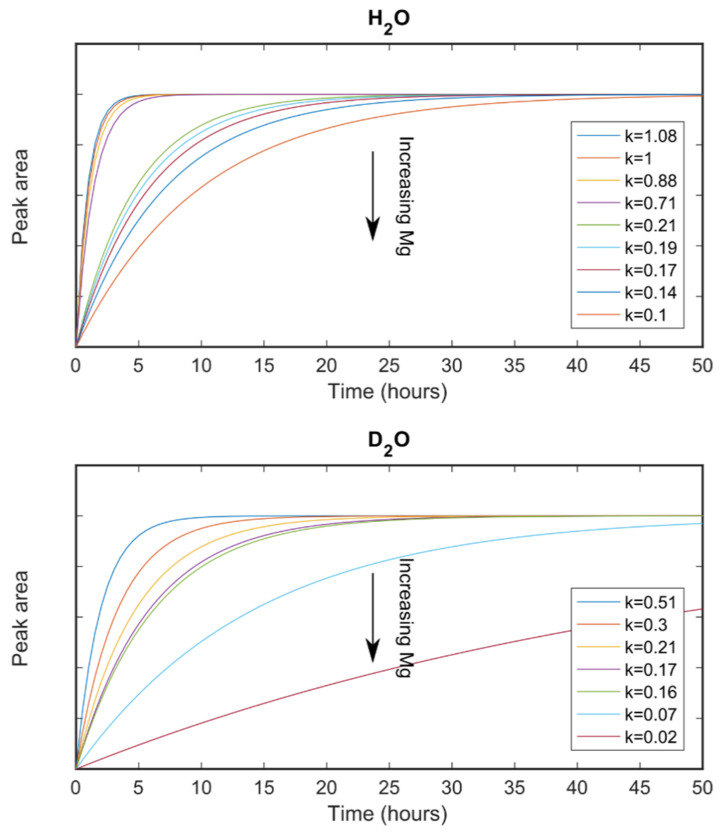
Compilation of the degree of aragonite formation as a function of magnesium content for samples formed from ACC with varying Mg/Ca ratios using H_2_O and D_2_O as the solvent.

**Figure 7 nanomaterials-12-04151-f007:**
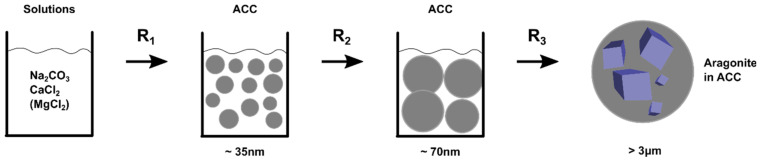
Schematic diagram summarizing the average particle pathway from solution to crystal for calcium carbonate, as observed in this work.

## Data Availability

All data is available on request from the corresponding author.

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
