# Peer review of "The Kinetics of Aragonite Formation from Solution via Amorphous Calcium Carbonate"

_nanomaterials, 2022, doi:10.3390/nano12234151_

Round 1

Author Response

Comment

Response

Sample preparation should be accurately described with all parameters (amount of solution,

concentration, mixing speed, etc.)

Done

Point 3.2.1 . It should be Raman spectroscopy instead of Raman spectrometry.

Done

Page 7. The reaction rate given in units hour -1 does not state the amount of crystals formed. The

content of the work shows that it was about the number of "of the initial 35 nm particles" created per

hour. It is also worth adding with a diameter of 35 nm.

Thanks for the comment.  We agree. The text in this section was confusing.  We have updated the text to clarify these points.

Page 8 lines 321-322. „leaving aragonite virtually free of Mg as is predicted from the aragonite

crystal chemistry.” Can you explain this statement? In other forms (including ACC), crystal chemistry

predicts the presence of Mg?

The text has been updated to clarify this point.

Page 8 lines 322-325. It is not clear? Does this mean that when dissolved, a new ACC is formed

with a higher content of Mg? What “Mg excluded from the aragonite lattice” means? Is aragonite

formed with Mg incorporated into the structure and is then dissolved?

Thanks for raising this point. The text is not clear.  We have modified the text to explain this more clearly.

Tab. 1. What does „H/D” mean?

The Table caption has been amended to clarify this point.

Parts 3.1.2, 3.2.2. X-ray diffraction. No data available in the manuscript.

Two extra figures containing the x-ray scattering from an ACC sample and the x-ray diffraction pattern of a sample crystallised from ACC have been included in the Supplementary Materials.  References to these figures have been included in parts 3.1.2 and 3.2.2.

Fig.1. why did the authors decide on such a range of Mg /Ca atomic ratio in solution? can we

expect to maintain a linear relationship at higher values?

This range has crystallisation rates that are fast enough and slow enough for us to be able to collect kinetic data. The text has been updated to make this clear.

Form Fig. S5 shows that ACC is formed on the aragonite surface. So where does the conclusion

about the occurrence of aragonite in the ACC come from?

The remanent ACC on the aragonite crystal surfaces is left over from the transformation and presumably contains high levels of Mg hence the long-lasting stability.  There is no aragonite in ACC, these are distinct chemical forms.  The text has been updated to clarify these points.

Has the presence of Mg been checked in the solution after crystallization to confirm its transition

from ACC to solution?

This measurement was not made during our study.  The Mg could go into solution or it could go into the ruminant ACC so making this measurement would not necessary answer the question anyway.  The text has been updated to include this point.

Reviewer 2 Report

Understanding the nucleation and growth of crystals is of great importance in the field of material science. This work by Simon et al demonstrated that Mg has a strong effect on crystal growth rate but little effect on the nucleation of crystal, which provided an insight into understanding the addicted Mg plays an exact role during the ACC transformation into aragonite. However, the lack of academic rigor especially in organizing the data reduces this work's reliability. Some important data need to be presented in the main text, but it is put into the SI or not shown, for example. I will give detailed comments below. Overall, I think this work advance understanding the of crystal formation via amorphous precursor and is suitable for publication in Nanomaterials after revision, providing the following issue being addressed.

1. Some experiments and data were described in the main text, but the experimental data not show. These data are a vital part of the whole study and also are supporting evidence for the academic conclusion. Therefore, these data should be provided in the main text. For example, the XRD data are not shown in sections 3.1.2 and 3.2.2; the Raman spectra are not shown in section 3.2.1;

2. SEM image is wrong identified by Figure S7, it should be Figure S5, see line 238. For the characterization of ACC particles, the TEM has a higher resolution and the SAED pattern also could confirm the crystal phase. These experiments are necessary to perform.

3. Figures 1, 2, 3, 4, and 5 with a low image resolution, please provide a higher resolution image.

4. The data shown in Figure S7 is important, which calculates the growth rate at various Mg/Ca ratios. Considering its significance for the whole story, it is better to present it in the main text. Similarly, Figure S9 also could be present in the main text.

5. Figure 5 describes the process of ACC growth and aragonite nucleation and growth. It just only provides a schematic diagram, it would be more clarity provide serial  TEM images with SAED patterns. 

Author Response

Comment

Response

1. Some experiments and data were described in the main text, but the experimental data not show. These data are a vital part of the whole study and also are supporting evidence for the academic conclusion. Therefore, these data should be provided in the main text. For example, the XRD data are not shown in sections 3.1.2 and 3.2.2; the Raman spectra are not shown in section 3.2.1;

The requested plots of the data have been added to the Supplementary Materials and referenced in sections 3.1.2, 3.2.2 and 3.2.1 as requested.

2. SEM image is wrong identified by Figure S7, it should be Figure S5, see line 238. For the characterization of ACC particles, the TEM has a higher resolution and the SAED pattern also could confirm the crystal phase. These experiments are necessary to perform.

Done.

3. Figures 1, 2, 3, 4, and 5 with a low image resolution, please provide a higher resolution image.

Separate files containing high resolution images for the figures in the main section have been provided.

4. The data shown in Figure S7 is important, which calculates the growth rate at various Mg/Ca ratios. Considering its significance for the whole story, it is better to present it in the main text. Similarly, Figure S9 also could be present in the main text.

Done. These figures have been moved into the main text as suggested.

5. Figure 5 describes the process of ACC growth and aragonite nucleation and growth. It just only provides a schematic diagram, it would be more clarity provide serial TEM images with SAED patterns. 

This is not a TEM study but a wide and small angle scattering study.  That is why there are no TEM images accompanying the manuscript.  Figure 5 is a summary of our interpretation of our wide and small angle scattering data.  Additional text has been added to the manuscript to clarify this point.

Reviewer 3 Report

Dear Authors!

Thank you for interesting investigation on the kinetic of aragonite formation from ACC!

The title of the article is clear and reflect the investigation content.

Abstract reflects the main points of the investigation

Keywords should be supplemented by materials: ACC, aragonite, and Ca/Mg ratio.

The introduction provides necessary state-of-art information and the goal of the investigation.

In methodology section, it should be noted that Mg/Ca ratio used instead of Ca/Mg in the abstract and the introduction. The Authors shoud use one type everywhere to improve the perception of the content.

Results:

Line 257: "We see" should be changed for "It was observed that"/"It can be seen that" etc. So as for line 259 ("we calculated") and elsewhere: impersonal constructions should be applied.   

Figures data are well-reported, but the quality of images should be improved.

Misprints:

Line 87, 245, 273: x in "x-ray" should be capitalised, so as m in "mg" (line 320).

References are formatted correctly, the level of self-citation is acceptable

Author Response

Comment

Response

Keywords should be supplemented by materials: ACC, aragonite, and Ca/Mg ratio.

Done

In methodology section, it should be noted that Mg/Ca ratio used instead of Ca/Mg in the abstract and the introduction. The Authors shoud use one type everywhere to improve the perception of the content.

Done

Line 257: "We see" should be changed for "It was observed that"/"It can be seen that" etc. So as for line 259 ("we calculated") and elsewhere: impersonal constructions should be applied.   

Done

Line 87, 245, 273: x in "x-ray" should be capitalised, so as m in "mg" (line 320).

The Chicago Manual of Style recommends x-ray is not capitalised as does IUPAC.  Mg has been corrected.

Round 2

Reviewer 1 Report

the authors correctly addressed all comments

Author Response

Thank you for the comment.

Reviewer 2 Report

The TEM observation of ACC particles should be performed, and relevant results need to be presented in the main text. Refer to comments 2 and 5.

Author Response

Thank you very much for the comments.  The team assembled for this work are experts in ACC and in x-ray and neutron scattering.  We are NOT experts in TEM studies.  The work we present is totally within our field of expertise.  We would NOT be comfortable presenting results in other fields of expertise.  The work presented here is a time resolved x-ray and neutron study of the kinetics of the formation of nano-crystalline ACC and subsequent nano-crystalline phases through to crystalline phases of Calcium/Magnesium Carbonate.  Although not a TEM expert I would struggle to see how a TEM study coud be used to measure the kinetics of this system.  Even if it could, we are not the team to do that and we leave that to other more qualified people.  The data here are completely sufficient for the conclusions drawn and we struggle to see how a TEM study could improve ths work in any way.